**Data Availability Statement:** To protect the identification of the patients, some restrictions do apply to the primary data. These data can be made

# Factors associated with community acquired severe pneumonia among under five children in Dhaka, Bangladesh: A case control analysis

Sabiha Nasrin[1], Md. Tariqujjaman[1], Marufa Sultana[1,2], Rifat A. Zaman[1], Shahjahan Ali[1], Mohammod J. Chisti[1], Abu S. G. Faruque[1]*, Tahmeed Ahmed[1], George J. Fuchs[3], Niklaus Gyr[4], Nur H. Alam[1]

**1** Nutrition and Clinical Services Division, icddr,b, Dhaka, Bangladesh, **2** Deakin Health Economics, School of Health and Social Development, Deakin University, Geelong, Australia, **3** Department of Pediatrics, College of Medicine and Department of Epidemiology, College of Public Health, University of Kentucky, Lexington, Kentucky, United States of America, **4** Faculty of Medicine, University of Basel, Basel, Switzerland

* gfaruque@icddrb.org

## Abstract

### Background

Pneumonia is the leading cause of death in children globally with the majority of these deaths observed in resource-limited settings. Globally, the annual incidence of clinical pneumonia in under-five children is approximately 152 million, mostly in the low- and middle-income countries. Of these, 8.7% progressed to severe pneumonia requiring hospitalization. However, data to predict children at the greatest risk to develop severe pneumonia from pneumonia are limited.

### Method

Secondary data analysis was performed after extracting relevant data from a prospective cluster randomized controlled clinical trial; children of either sex, aged two months to five years with pneumonia or severe pneumonia acquired in the community were enrolled over a period of three years in 16 clusters in urban Dhaka city.

### Results

The analysis comprised of 2,597 children aged 2–59 months. Of these, 904 and 1693 were categorized as pneumonia (controls) and severe pneumonia (cases), respectively based on WHO criteria. The median age of children was 9.2 months (inter quartile range, 5.1–17.1) and 1,576 (60%) were male. After adjustment for covariates, children with temperature ≥38˚C, duration of illness ≥3 days, male sex, received prior medical care and severe stunting showed a significantly increased likelihood of developing severe pneumonia compared to those with pneumonia. Severe pneumonia in children occurred more often in older children who presented commonly from wealthy quintile families, and who often sought care from private facilities in urban settings.

available from the Ethics Committees (ERC/RRC) at the International Centre for Diarrhoeal Disease Research, Bangladesh (icddr,b) for researchers who meet the criteria for access to confidential data; please contact the Head of Research Administration at the icddr,b (Armana Ahmed; aahmed@ICDDRB.org.)

**Funding:** NHA received all the funding's UBS Optimus Foundation Switzerland Unicef -Bangladesh and Switzerland Foundation Botnar, Switzerland Eagle Foundation, Switzerland The funders had no role in study design, data collection and analysis, decision to publish, or preparation of the manuscript.

**Competing interests:** The authors have declared that no competing interests exist.

## Conclusion and recommendation

Male sex, longer duration of illness, fever, received prior medical care, and severe stunting were significantly associated with development of WHO-defined severe childhood pneumonia in our population. The results of this study may help to develop interventions target to reduce childhood morbidity and mortality of children suffering from severe pneumonia.

## Introduction

Globally, pneumonia is the leading cause of death among under-five children with more than 90% of these occurring in resource-limited settings [1]. Implementation of feasible and effective interventions has reduced under-five pneumonia death substantially from 13·6 per 1000 livebirths in 2000 to 6·6 per 1000 livebirths in 2015 [2, 3], yet, pneumonia ranked top in mortality. Childhood death due to pneumonia occurs disproportionately in low-and middle-income countries (LMICs) with the greatest number observed in South Asia and sub-Saharan African countries [3, 4]. Compared to deaths due to other childhood diseases, pneumonia related deaths are declining at a slower rate [5]. Rudan et al. reported the incidence of clinical pneumonia in under-five children was approximately 0.29 episodes per child-year in LMICs of the world [6]. This means, every year 151.8 million incidences with 13.1 million (8.7%) episodes progressing from pneumonia to severe pneumonia that require hospitalization [6, 7].

According to World Health Organization (WHO), United Nations Children's Fund (UNICEF), and others, risk factors for severe childhood pneumonia in LMICs include bacterial etiology, young age, low birth weight, malnutrition, household crowding, exposure to indoor air pollution, and low-level schooling of mothers [8–10]. Factors associated with severe pneumonia in different contexts and their changes over time in LMICs are incompletely defined [6, 11, 12]. In Bangladesh, severe pneumonia is the main cause of hospitalization among under-five children. Among 156,847 admitted children identified in a surveillance performed in seven hospitals in Bangladesh from May 2004 through April 2007, the most frequent diagnosis was pneumonia (32%) [10].

We analysed a large dataset of children with severe pneumonia/pneumonia from an urban facility-based study in Bangladesh to explore factors associated with severe pneumonia compared to pneumonia. We included clinical, socio-demographic characteristics, and care-seeking behaviour of the study population in analyses to identify the associated factors of severe pneumonia among under-five children.

## Materials and methods

### Study setting

This study was conducted in sixteen urban Wards (lowest administrative unit of City Corporation of Dhaka, the capital city of Bangladesh, with an average population of 200,000), between January 2016 and March 2019. The study was implemented in the Smiling Sun Franchise clinics (SH clinics) run by Non-governmental organizations (NGOs) in collaboration with the Dhaka City Corporation and financially supported of the donors [United States Agency for International Development (USAID) and Foreign, Commonwealth & Development Office (FCDO)]. Children were either self-referred or referred from other facilities within the city. One group received treatment on a day care basis (Day Care Approach, DCA) for severe

pneumonia and the other group received usual standard treatment in one of various hospitals in Dhaka city based on cluster randomization.

## Study design

Secondary data analysis was performed following a case-control design after extracting data from the database of the study entitled "Effectiveness and safety of Day Care versus Usual Care Management of Severe Pneumonia with or without Malnutrition in Children Using the Existing Health System of Government of Bangladesh". The main study was a prospective, cluster randomized controlled clinical trial (registered at www.ClinicalTrials.gov Identifier: NCT02669654). Children presenting to study clinics were assessed and those diagnosed as severe pneumonia defined by WHO criteria [13] were considered as cases and those with pneumonia served as analyzable controls. The Institutional Review Board (IRB) of icddr,b, including the Research Review Committee (RRC) and Ethical Review Committee (ERC), approved the study.

## Sample size

The secondary data analysis was done with the available samples in the study. Children presenting to study clinics were assessed and those diagnosed as severe pneumonia, N = 1963, and pneumonia, N = 904 were included in this analysis.

## Operational definitions

Pneumonia was defined as a history of cough or difficult breathing and lower chest wall indrawing or age-specific fast breathing ($\geq$50 and $\geq$40 breaths/minute for 2–11 month-olds and 12–59 month-olds respectively) without any general danger signs [13, 14]. Severe pneumonia was defined as pneumonia with at least one of the following danger signs: central cyanosis or hypoxemia (oxygen saturation < 90% measured by pulse oximeter), severe respiratory distress (e.g. grunting, very severe chest in-drawing), inability to breastfeed or drink, lethargy or unconscious, or convulsion.

Nutritional status was characterized as moderate acute malnutrition (weight-for-height/ length Z-score $\geq$-3 ZWH and $\leq$-2 or Mid-upper arm circumference (MUAC) $\geq$ 115 mm and < 125 mm for children aged 6–59 months); severe acute malnutrition (<-3 weight-for-height/length Z-score or with nutritional edema, or MUAC <115 mm, either alone or in combination); severe underweight (weight for age Z score <-3 ZWA); severe stunting (height/length-for-age Z score <-3 HAZ) [15].

Exclusive breastfeeding was defined as the child received only breast milk with no other liquids or solids provided (not even water) with the exception of oral rehydration solution, or drops/syrups of vitamins, minerals or medicines for the first 6 months of life. Thereafter, infants receive complementary foods with continued breastfeeding up to 2 years of age or beyond [16].

Hospital-acquired pneumonia (an infection occurring in a patient during the process of care in a hospital or other health care facility which was not present or incubating at the time of admission); bronchiolitis (diagnosed by single-dose bronchodilator challenge test); suspected sepsis (very ill, fever, altered mentation, convulsion); meningitis (very irritable, stiff neck, petechial rash); diarrhea (3 or more stools per 24 hours); severe dehydration (lethargic, sunken eyes, skin pinch goes back slowly, unable to drink); bronchial asthma (history of recurrent wheezing, responded to bronchodilator, prolonged expiration), above disease conditions according to WHO guideline [13, 17, 18].

## Data collection

The study cohort included children of either sex, aged 2–59 months with clinical baseline diagnosis of pneumonia (controls) or severe pneumonia (cases) in the triage area by obtaining written informed consent from parents or legal guardians. The children sought care for severe pneumonia with or without moderate acute malnutrition (MAM), severe underweight, and diarrhoea with no or some dehydration. Children were excluded if observed to have any of the following: hospital-acquired pneumonia, severe acute malnutrition, bronchiolitis, suspected sepsis, meningitis, convulsion, congenital heart disease, diarrhoea with severe dehydration, bronchial asthma or reporting with any other associated life-threatening illness.

Standard structured questionnaires consisting of basic socio-demographic, clinical, health-care seeking behaviour, and associated variables of study interest (see data analysis section) were completed. Mothers were encouraged to bring their child's National Immunization Card so that research staff could abstract information on the exact age and immunization status of the child.

## Case management

At DCA clinics, a study physician evaluated children for the presence of pneumonia/severe pneumonia with or without moderate acute malnutrition. Children who met inclusion criteria and identified as pneumonia were treated at home with oral antibiotic (syrup Amoxicillin) for 5 days. Those who failed after two days of oral antibiotic treatment were enrolled as severe pneumonia (WHO does not categorize them as severe pneumonia but recommend prompt referral to a facility available with second-line management, to develop a simplified approach that could increase the number of children receiving appropriate management for pneumonia). At the same time, children with severe pneumonia on presentation were also enrolled and treated either at a DCA clinic or hospital. The children admitted in the hospital either by self-referral or referred by our study personnel or any physician in the community, received usual treatment according to the respective hospital treatment protocol.

## Data analysis

Data were analysed using STATA (StataCorp version 13) and analyses included descriptive as well as analytic methods. Frequencies with percentages for categorical variables, medians with interquartile ranges (IQR) for continuous skewed variables, and means with standard deviations for normally distributed continuous variables were calculated to summarize the data. The independent variables were analysed in the simple binary logistic regression model, and the attributes that were observed to be significantly associated (p-value <0.05) with the dependent variable (severe pneumonia) along with clinically relevant non-significant but important variables of public health interest were included in the multivariable logistic regression model. The strength of association between outcome variable and the independent variables of interest were assessed by calculating Odds ratios (ORs) with 95% confidence intervals. We checked whether or not the model fitted well by the goodness of fit test and ROC curve. We also checked the muticollinearity among the independent variables by the variance inflation factor (VIF).

## Dependent and independent variables

Pneumonia severity was the dependent variable and its options were: severe pneumonia = 1 and pneumonia = 0. The independent variables included in this study were; Associated *clinical features*: duration of illness before enrolment, temperature, pulse rate, MUAC (in cm), weight-

for-age Z score (WAZ), weight-for-height/length Z score (WHZ), height-for-age Z score (HAZ), and associated comorbidity (diarrhoea). *Socio-economic-demographic factors*: age (infants 2–11 months, young children 12–23 months, toddlers 24–59 months), sex, exclusively breastfed, immunization status, father's education, mother's/caregivers education, number of household members, number of siblings, household monthly income, wealth index, type of delivery of the baby at birth. *Health care seeking behaviours*: reporting to facility or health care provider prior to study enrolment. *Environmental factors*: predominant structure of wall in the house, floor material, water treatment method, and cooking fuel source.

Categorical independent variables were coded as: duration of illness (<3days = 0, ≥3 days = 1); temperature (temperature ≥38˚C = 1, temperature <38˚C = 0); male child (male = 1, female = 0); age (infants 2–11 months = 2, children 12–23 months = 1, children 24–59 months = 0); exclusively breastfed (yes = 0, no = 1); Pentavalent vaccine recipient (yes = 0, no = 1); PCV vaccine recipient (yes = 0, no = 1); measles vaccine recipient (yes = 0, no = 1); father's education (Illiterate = 3, Primary = 2, Secondary = 1, Higher = 0); mother's/caregiver's education (Illiterate = 3, Primary = 2, Secondary = 1, Higher = 0); household member (<5 person = 0, ≥5 person = 1); number of siblings (two or more = 1, one = 0); type of delivery (normal vaginal delivery = 0, caesarean section = 1); received treatment prior to enrolment (private = 0, public = 1, pharmacy = 2, others = 3); consume safe drinking water (filter/boil/chlorine tablet = 0, no = 1); use improved toilet facility (flush/ventilated pit improved = 0, not improved = 1); wealth index (wealthiest = 0, wealthier = 1, middle = 2, poorer = 3, poorest = 4); Predominant wall in the house (brick = 0, others = 1); floor materials (cement/ceramic tiles = 0, others = 1); gas fuel(yes = 1, no = 0); wood (yes = 1, no = 0); underweight status (severe = 1, moderate + normal = 0); stunting status (severe = 1, moderate + normal = 0); and diarrhoea (yes = 1, no = 0).

## Results

The study comprised of 2597 children aged 2–59 months for the analysis. Of these, 1693 were cases and 904 were controls (Fig 1).

The median age of study children was 9.2 months (IQR, 5.1–17.1), 1576 (60%) were male, and 1534 (59%) were in the 2-11-month age group. Two thirds of study children had more than two siblings. Cases often had higher fever (41% vs 20%), and higher pulse rate (17% vs 16%) compared to controls. Hypoxemia was only observed in cases (11% of cases). Regarding

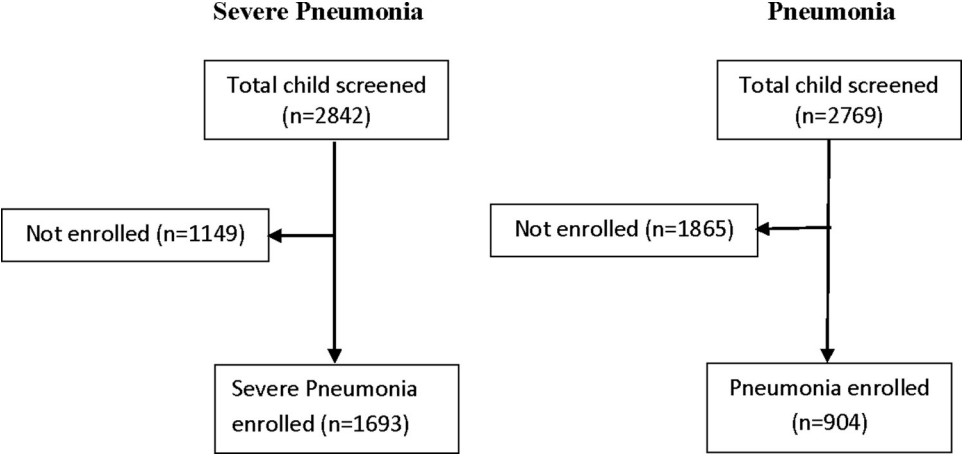

**Fig 1. Enrolment profile of the study participants.**

socio-demographic, care seeking and environmental variables; we also observed differences in child's sex, household income, received treatment prior to enrolment, child's mode of delivery during birth, treatment method of drinking water, toilet facility, housing (wall, floor), cooking source (gas, wood). Care seeking behaviours were observed nearly two times more frequent in cases than in their control counterparts (73% vs. 43%). Similarly, monthly household income was higher in cases compared to controls. Further, we found minimal differences between the groups in immunization status, nutritional status, household size, father's education, or presence of diarrhoea (Table 1).

**Table 1. Comparison of characteristics of the study population with severe pneumonia (cases) and pneumonia (controls) according to the WHO classification1.**

| Variables | Severe Pneumonia n = 1693 (%) | Pneumonia n = 904(%) |
|---|---|---|
| **Presenting clinical features** | | |
| Duration of illness > 3days | 1536 (67.2) | 751 (32.8) |
| Temperature $\geq$ 38˚C | 701 (41.4) | 176 (19.5) |
| Pulse rate/min, mean (SD) [b] | 140.9 (16.8) | 139.0 (15.8) |
| **Nutritional status** | | |
| MUAC[a], mean (SD) | 13.7 (1.2) | 14.0 (1.2) |
| Severe underweight (<-3SD) | 125 (7.4) | 55 (6.2) |
| Severe stunting (<-3SD) | 221 (13.1) | 97 (10.9) |
| **Sociodemographic profile** | | |
| Gender (male) | 1066 (63.0) | 510 (56.4) |
| **Child's age** | | |
| 2–11 mo. | 1040 (61.4) | 494 (54.7) |
| 12–23 mo. | 408 (24.1) | 255 (28.2) |
| 23–59 mo. | 245(14.5) | 155 (17.6) |
| **Exclusive breastfed** | 858 (50.7) | 490 (54.2) |
| **Immunization status** | | |
| Pentavalent | 1599 (94.5) | 855 (94.6) |
| PCV | 1466 (86.6) | 759 (84.0) |
| Measles | 211/439 (48.1) | 336/644 (52.8) |
| **Father's education** | | |
| Illiterate | 252 (15.0) | 157 (18.1) |
| Primary | 483 (28.7) | 254 (29.3) |
| Secondary | 677 (40.2) | 311 (35.9) |
| Higher | 274 (16.3) | 144 (16.6) |
| **Mother's/caregiver's education** | | |
| Illiterate | 189 (11.2) | 131 (14.6) |
| Primary | 522 (30.9) | 321 (35.9) |
| Secondary | 773 (45.7) | 338 (37.8) |
| Higher | 207 (12.2) | 105 (11.7) |
| **Household member** | | |
| $\geq$5 | 829 (49.0) | 436 (48.3) |
| **Number of siblings** | | |
| Two or more | 1020 (60.3) | 549 (60.8) |
| **Socio-economic context** | | |
| **Household income in BDT, Median (IQR) [c]** | 19000 (14000, 30000) | 15000 (12000, 24000) |
| **Wealth index** | | |
| Wealthiest | 383 (22.6) | 125 (13.9) |

*(Continued)*

**Table 1.** (Continued)

| Variables | Severe Pneumonia n = 1693 (%) | Pneumonia n = 904(%) |
|---|---|---|
| Wealthier | 374 (22.1) | 154 (17.1) |
| Middle | 326 (19.3) | 181 (20.1) |
| Poorer | 312 (18.4) | 215 (23.9) |
| Poorest | 297 (17.6) | 225 (25.0) |
| **Received care** | | |
| Private | 492 (39.2) | 97 (25.1) |
| Public | 156 (12.4) | 21 (5.4) |
| Pharmacy | 555 (44.3) | 244 (63.2) |
| Others | 51 (4.1) | 24 (6.2) |
| **Type of delivery** | | |
| Caesarean section | 752 (44.4) | 356 (39.4) |
| **Environment and household hygiene** | | |
| Treatment of Drinking water, safe | 1036 (61.2) | 467 (51.7) |
| Toilet Facility, improved[d] | 1019 (60.2) | 411 (45.5) |
| **Predominant wall in the house** | | |
| Brick | 1347 (79.6) | 639 (70.7) |
| **Floor materials** | | |
| Cement | 1430 (84.5) | 734 (81.1) |
| **Cooking fuel source** | | |
| Wood | 182 (10.8) | 139 (15.4) |
| Gas | 1557 (92.0) | 777 (86.1) |
| **Co-morbidities** | | |
| Diarrhoea | 65 (3.8) | 24 (2.7) |

[a]MUAC = Mid Upper Arm Circumference in cm

[b]SD = Standard Deviation

[c] IQR = Inter Quartile Range

[d] improved = flush+ pour flush+ ventilated pit latrine

[1]Revised WHO classification and treatment of childhood pneumonia at health facilities 2014

In unadjusted analysis, factors significantly associated with severe pneumonia included longer duration of illness, temperature $\geq 38°$C, increased pulse rate/min, male sex, 2–11 months old age group, wealth index, received treatment prior to enrolment, child's mode of delivery during birth, housing (wall, floor), cooking source (gas, wood), drinking water treatment method, and toilet facility (Table 2).

Factors significantly associated with cases after adjusting for duration of illness, temperature, pulse rate, child's age, child's sex, stunting (nutritional status), received prior medical care, wealth index, mother's education, cooking fuel source (wood, gas) in logistic regression analysis included: duration of illness $\geq 3$days, temperature $\geq 38°$C, severe stunting, male sex, wealth index, received prior medical care. Most children from the wealthiest quintile (60%) received care from private or public facilities whereas those from the poorest quintile (70%) sought care from pharmacies/drug stores and other facilities (data not presented). Cases sought care more often from appropriate facilities compared to the control group. In the final multivariable model, Hosmer-Lemeshow goodness-of-fit test showed it to be non-significant (p- value = 0.26) indicating the model fitted well. The VIF values of all independent variables were 2.81 or less and the mean VIF was 1.79. The value under the ROC curve was 0.7483. We have also tested for interaction. No effect modification was observed.

**Table 2. Factors associated with severe pneumonia compared to pneumonia among under-five children.**

| Risk factors | Unadjusted OR (95% CI) | p-value | *Adjusted OR (95% CI) | p-value |
|---|---|---|---|---|
| **Presenting clinical features** | | | | |
| **Duration of illness** | | | | |
| <3 days | Reference | | Reference | |
| ≥3 days | 1.13 (1.09, 1.16) | <0.001 | 1.55 (1.19, 2.02) | 0.001 |
| **Temperature ≥ 38˚C** | | | | |
| No | Reference | | Reference | |
| Yes | 2.92 (2.41, 3.54) | <0.001 | 2.66 (2.17, 3.26) | <0.001 |
| **Severe stunting** | | | | |
| No | Reference | | Reference | |
| Yes | 1.23 (0.95, 1.58) | 0.109 | 1.44 (1.09, 1.91) | 0.009 |
| **Sociodemographic profiles** | | | | |
| **Child's sex** | | | | |
| Female | Reference | | Reference | |
| Male | 1.31 (1.11, 1.55) | 0.001 | 1.33 (1.11, 1.60) | 0.002 |
| **Wealth index** | | | | |
| Wealthiest | Reference | <0.001 | Reference | <0.001 |
| Wealthier | 0.79 (0.6, 1.04) | | 0.58 (0.41, 0.83) | |
| Middle | 0.58 (0.44, 0.77) | | 0.49 (0.34, 0.70) | |
| Poorer | 0.47 (0.36, 0.61) | | 0.41 (0.28, 0.59) | |
| Poorest | 0.43 (0.33, 0.56) | | 0.47 (0.31, 0.70) | |
| **Received care** | | | | |
| Private | Reference | <0.001 | Reference | <0.001 |
| Public | 1.46 (0.88, 2.42) | | 1.28 (0.70, 2.33) | |
| Pharmacy | 0.45 (0.34, 0.58) | | 0.47 (0.34, 0.65) | |
| Others | 0.18 (0.14, 0.23) | | 0.22 (0.16, 0.29) | |

Abbreviations: OR = Odds Ratio, CI = Confidence Interval

*Adjusted for temperature, duration of illness, pulse rate, child's age, child's sex, stunting, received prior medical care, wealth index, mother's education, cooking fuel source- gas, wood

Results of unadjusted and adjusted odds ratio were calculated using simple and multivariable logistic regression

## Discussion

Currently, in-depth understanding of childhood pneumonia based on research is greatest in context of high-income countries compared to LMICs. Definition of epidemiology of the illness, causal pathogens, and key prognostic factors is more robust in such settings, many of which are different in LMICs. According to available literature, knowledge of factors and especially clinical risk factors associated with severe paediatric pneumonia at the country as well community level in LMICs is limited [19].

Male children were predominant in the severe pneumonia group (63%) in our study. Yet in the pneumonia group their proportion reached 56% as well; overall male children thus were more affected significantly by severe pneumonia corroborating findings of other studies [7, 8, 20]. The cause behind the high susceptibility of male children could be either genetic, or higher reporting for male children by the mothers due to gender bias, which potentially causes mothers to notice symptoms due to a higher attention to male children particularly for seeking health care much earlier than female children [21]. Nevertheless, boys have greater likelihood of being affected or of care seeking in general for common acute respiratory illness than girls, as reported in several studies from Bangladesh [22–24]. Another possibility of male children to

be in the high risk of infection could be the testosterone suppressing the immune response [23]. However, clarification of this trend is multifaceted, as the role of social determinants of health, such as sex, socio-economic status (SES), and water, sanitation, and hygiene (WASH) practices, with disease are often not included in studies.

For safety reasons children with severe acute malnutrition were excluded from the main effectiveness trial although children with moderate malnutrition were included as mentioned earlier. One systematic review suggested that like the severely malnourished children, children with moderate degrees of malnutrition may also be at increased the risk of death due to pneumonia [25, 26]. Studies that evaluated the impact of moderate malnutrition are comparatively few as moderate degree of malnutrition in health facilities in developing countries is not recorded as an admission diagnosis. Notably, we found severe stunting to be associated with severe pneumonia. Stunting is established to have long-term sequels on lung development and growth and to be associated with prolonged acute course of pneumonia treatment and delay in recovery [27]. Malnutrition in children results in an immunocompromised state and subsequent increase in infectious morbidity and mortality due to impairments in multiple aspects of the immune system including cell mediated and complement responses, inefficient chemotaxis, reduced mature T cells, compromised phagocytic activity, among others. The results of our study highlight the importance of stunting in children being treated for severe pneumonia [28, 29].

Hypoxaemia, a major indicator of disease severity, was observed in almost 11% children with severe pneumonia by pulse oximeter, consistent with findings from a systematic review of childhood pneumonia in LMICs [7]. The need for application of pulse oximeter and related staff training to accurately identify and monitor children with hypoxaemia is therefore self-evident [7, 30].

On the World Pneumonia Day on Nov 12, 2015, action was sought to improve the early identification and treatment of childhood pneumonia at community and outpatient level to reduce disease severity and deadly outcomes [31]. It was apparent that the case–mortality rate in untreated children with pneumonia is high, sometimes reaching as high as 20%, and deaths can occur as early as 3 days after illness onset [31]. We also found that duration of illness at home for 3 days or more was significantly associated with the likelihood of disease progression to severe pneumonia. The same observation was reported from Kenya [32]. In our study fever was significantly associated with severe pneumonia. Studies in diverse LMICs like South Africa, Papua New Guinea, and Indonesia reported no association between fever and pneumonia severity [19, 33–35]. However, one study in USA indicated temperature to be associated with severe pneumonia (not defined by WHO classification) [36] and another study reported duration of fever (at day 6) was associated with severity [37]. Although WHO did not consider fever in their pneumonia severity criteria, the British Thoracic Society (BTS) includes fever in their guideline [13, 38].

This study also depicted a noteworthy association of care-seeking behaviour with socio-economic status variables which may be new information. Although, few reports are available on severe pneumonia and those two variables according to most literature on children with pneumonia. We found care seeking behaviour proportion to be higher in much wealthier cases compared to controls. Overall 62.3% households looked for care from different health care facilities prior to enrolment in our study. Parents from the highest wealth index quintile were observed to seek care from qualified providers (in private or public facilities). These observations have also been confirmed by earlier studies [39, 40]. Not surprisingly, increased awareness and ability to meet the expense of care play a remarkable role in decision making, as people from higher wealth quintiles are often more aware and generally able to afford health care cost [40]. In contrast to the wealthiest group, the poorest groups sought health care from

the pharmacies and other nearby facilities including the traditional healers. This pattern of behaviour has been reported in other recent studies [40, 41]. However, severe outcome of pneumonia and increased mortality rates have consistently been associated with low compared with high SES [42]. The Integrated Management of Childhood Illness (IMCI) guidelines [43] suggested awareness raising interventions to improve family and community practices through the education of mothers, fathers, other child caretakers, and members of the community, with a focus on timely care seeking from appropriate facility, compliance, initial care at home, and overall health promotion. This finding is difficult to explain may be more resources and information are available in the richest quintile to access nearby facilities. They could be much aware and often presented to the appropriate facility in urban area.

Exclusive breastfeeding, parent's education, immunization, household size, number of siblings, housing type, and cooking fuel (gas) were not significantly associated with severe pneumonia in our study. Nevertheless, several studies showed some of these factors being significantly associated with pneumonia/severe pneumonia [8, 19, 44]. A possible explanation could be our participant's baseline admission characteristics were identical in both the groups and thus validate the study results. The majority of the urban households used gas stove for cooking purposes and have same type of housing. A comparison of urban with rural data might possibly could reveal variation in housing type, cooking fuel use, etc. Pentavalent vaccine and PCV coverage were 90% in our patient population, known interventions that reduce severe bacterial pneumonia [45]. In Bangladesh we have higher vaccine coverage (>90%), thus we are in the right track in terms of our "Expanded Programme of Immunization" coverage [45, 46]. No significant association was found between diarrhoea as comorbidity and the risk of severe pneumonia. However, in our study population, only a small proportion of patients presented with diarrhoea as co-morbidity.

### Strengths of the study

Data were collected over few years which should have had captured the seasonal variations. Clinical data were taken instantly. This study had a strong referral backup system for DCA admitted children and study sponsors with a sustained commitment over an extended period of time that enabled what would have otherwise been a challenging study to conduct at the community level. We captured heterogeneity (clinical, socio-demographic, care seeking and environmental variables) in patient characteristics; our model explains 74% predicted probabilities between outcome and predictors and we, therefore, conclude that our model fits well.

### Limitations of the study

Our study was not designed to determine factors associated with population-based pneumonia and severe pneumonia children; only the number and type of cases treated in the urban site were included in this analysis. For ethical reasons we included only pneumonia or severe pneumonia with or without moderate malnutrition in our study and excluded high mortality-related SAM children with severe pneumonia. Lastly, the selection of cases and control was subjective as per the WHO algorithm and we performed a limited number of chest X-rays or other laboratory testing to confirm diagnosis due to attempts of cost containment so there was a chance of misclassification.

### Conclusion

This study analysed a large number of clinical pneumonia episodes among under-five children in Bangladesh. We found male children, longer duration of illness, fever, received prior medical care, and severe stunting as significantly associated factors for severe pneumonia compared

to pneumonia. The results of this study have the potential to help refine decisions about case management in resource-limited countries by facilitating decisions about the most appropriate site of treatment (i.e. home vs. hospital) or the need for additional supportive care. There is a pressing need for further research in larger populations to further define significant risk factors of severe pneumonia from the community and address key knowledge gaps in order to enable optimal management strategies with the potential for substantial reductions in morbidity and mortality.

## Acknowledgments

We would like to thank study participants, data collectors, and supervisors for their unreserved contribution during data collection. We gratefully acknowledge our core donors for their unrestricted support and commitment to icddr,b's research efforts include the Governments of Bangladesh, Canada, Sweden and the UK.

## Author Contributions

**Conceptualization:** Sabiha Nasrin, Abu S. G. Faruque, Nur H. Alam.

**Data curation:** Sabiha Nasrin, Md. Tariqujjaman.

**Formal analysis:** Sabiha Nasrin, Md. Tariqujjaman.

**Funding acquisition:** Nur H. Alam.

**Methodology:** Abu S. G. Faruque, Nur H. Alam.

**Project administration:** Shahjahan Ali.

**Resources:** Sabiha Nasrin.

**Supervision:** Mohammod J. Chisti, Abu S. G. Faruque, Tahmeed Ahmed, George J. Fuchs, Niklaus Gyr, Nur H. Alam.

**Writing – original draft:** Sabiha Nasrin.

**Writing – review & editing:** Md. Tariqujjaman, Marufa Sultana, Rifat A. Zaman, Mohammod J. Chisti, Abu S. G. Faruque, Tahmeed Ahmed, George J. Fuchs, Niklaus Gyr, Nur H. Alam.

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
