## [Decision Letter · Decision Letter 0]

3 Nov 2021

PONE-D-21-06226Factors associated with community acquired severe pneumonia among under five children in Dhaka, Bangladesh: A case control studyPLOS ONE

Dear Dr. Faruque,

Thank you for submitting your manuscript to PLOS ONE. After careful consideration, we feel that it has merit but does not fully meet PLOS ONE’s publication criteria as it currently stands. Therefore, we invite you to submit a revised version of the manuscript that addresses the points raised during the review process. The manuscript has been evaluated by two reviewers, and their comments are available below. The reviewers have raised a number of concerns that need attention. In particular, reviewer #1 has raised several points regarding the statistical analysis and the general reporting of the methodology, including a specific request to follow the STROBE guidelines. Could you please revise the manuscript to carefully address the concerns raised?

We look forward to receiving your revised manuscript.

Kind regards,

Dario Ummarino, Ph.D.

Senior Editor

PLOS ONE

Journal Requirements:

Reviewers' comments:

Reviewer's Responses to Questions

**Comments to the Author**

1. Is the manuscript technically sound, and do the data support the conclusions?

Reviewer #1: Yes

Reviewer #2: Yes

2. Has the statistical analysis been performed appropriately and rigorously? 

Reviewer #1: No

Reviewer #2: Yes

3. Have the authors made all data underlying the findings in their manuscript fully available?

Reviewer #1: No

Reviewer #2: Yes

4. Is the manuscript presented in an intelligible fashion and written in standard English?

Reviewer #1: Yes

Reviewer #2: Yes

5. Review Comments to the Author

Reviewer #1: Abstract: 'annual incidence' in line 22 and 'each year' on line 23 are a repetition.

Introduction

- since you are reporting absolute incidences, it would be useful to also indicate the global change in the population of the children referred to in line 49 (i.e. the denominator for those incidences).

- would it be accurate to use 'non-severe pneumonia' to refer to what you currently call 'pneumonia'; for example, in line 70: "we analysed a large dataset of children with severe and non-severe pneumonia..." and in line 71/72: "... factors associated with severe pneumonia compared to non-severe pneumonia."

Methods

- please report the methods according to STROBE guidelines for case-control studies (see https://www.equator-network.org/wp-content/uploads/2015/10/STROBE_checklist_v4_case-control.pdf).

- there doesn't seem to be any value on the chi-squared tests, Mann-Whitney tests or t-tests listed in line 175 given your intention to conduct univariable regression models as described in line 177.

Results

- linked to the last comment in methods (above), the results described in line 246 relating to unadjusted regression analysis would be expected to be identical to those of the univariate test p-values in Table 1. I would recommend that you keep table 1 purely descriptive with no hypothesis tests of the differences between the two groups and p-values, as this is what table 2 explores.

- in table 2, you should report a single likelihood ratio p-value or global wald test for multicategorical variables such as wealth index and received care, not the individual wald test p-values comparing each category to the reference category as currently done. It is that single p-value that should be used to assess evidence of association with the whole variable. For example, you report that received care was no longer associated with the outcome after adjusting for other variables, but this is unlikely to be correct based on a single global test (as described above).

Reviewer #2: The presence of crepitation indicating alveolar involvement is part of the clinical diagnosis of pneumonia. It is possible that its absence in the control group may have occurred due to differential diagnoses of other respiratory diseases. I suggest removing the presence of crepitation as a risk factor.

Regarding size, it can be as the authors did, although in our opinion it would be more appropriate to respect the case-control (1:1 ratio) study design and the cases and controls had the same number of participants.

6. PLOS authors have the option to publish the peer review history of their article (what does this mean?). If published, this will include your full peer review and any attached files.

Reviewer #1: No

Reviewer #2: **Yes: **Eduardo Jorge da Fonseca Lima

---

## [Author Response · Author response to Decision Letter 0]

11 Dec 2021

Reviewer #1: 

Comment:

Abstract: 'annual incidence' in line 22 and 'each year' on line 23 are a repetition.

Response: Thank you very much for spotting this. We have deleted “each year” from line 24 as suggested. 

Comment:

Introduction

- since you are reporting absolute incidences, it would be useful to also indicate the global change in the population of the children referred to in line 49 (i.e. the denominator for those incidences).

Response: Thank you so much for the suggestion. We have edited line 49, it now reads “Implementation of feasible and effective interventions has reduced under-five pneumonia death substantially from 13·6 per 1000 livebirths in 2000 to 6·6 per 1000 livebirths in 2015” (line 61).

Comment:

- would it be accurate to use 'non-severe pneumonia' to refer to what you currently call 'pneumonia'; for example, in line 70: "we analysed a large dataset of children with severe and non-severe pneumonia..." and in line 71/72: "... factors associated with severe pneumonia compared to non-severe pneumonia."

Response: We appreciate and accept your suggestion. We have replaced all “non-severe pneumonia” term to “pneumonia” (line 112, 258). 

Comment: 

Methods

- please report the methods according to STROBE guidelines for case-control studies (see https://www.equator-network.org/wp-content/uploads/2015/10/STROBE_checklist_v4_case-control.pdf).

Comment: Thank you so much for the excellent suggestion. Please find the STROBE table below. We have mentioned the line numbers where the information was specified in the manuscript. Find the full checklist in "Reviewers Response" document, but a summary is given below:

Title and abstract Line 1 and 19

Background/rationale Lines 50-80

Objectives Lines 82-86 

Study design Lines 105-112

Setting Lines 95-101

Participants Lines 134-157

Outcomes, exposures: Lines 195-223

Data sources Lines 105-112

Study size Lines 111-112

Statistical methods Lines 181-193

Results Line 230 (Fig 1.)

Descriptive data Lines 245-250;Table 1.

Outcome data Lines 258-264

Comment:

- there doesn't seem to be any value on the chi-squared tests, Mann-Whitney tests or t-tests listed in line 175 given your intention to conduct univariable regression models as described in line 177.

Response: Thank you for your comment. To get the p-values in table 1, We used the chi-squared tests, Mann-Whitney tests, or t-tests. However, as per your other suggestion, we are keeping Table 1 pure descriptive, so, we have deleted the line from data analysis section.

Comment:

Results

- linked to the last comment in methods (above), the results described in line 246 relating to unadjusted regression analysis would be expected to be identical to those of the univariate test p-values in Table 1. I would recommend that you keep table 1 purely descriptive with no hypothesis tests of the differences between the two groups and p-values, as this is what table 2 explores.

Response: Authors agree with the suggestion. We have edited Table 1 and kept it purely descriptive as per your valuable suggestion. (Line 245).

Comment:

- in table 2, you should report a single likelihood ratio p-value or global wald test for multicategorical variables such as wealth index and received care, not the individual wald test p-values comparing each category to the reference category as currently done. It is that single p-value that should be used to assess evidence of association with the whole variable. For example, you report that received care was no longer associated with the outcome after adjusting for other variables, but this is unlikely to be correct based on a single global test (as described above).

Response: We appreciate your suggestion. We re-analyzed the data according to your suggestion and presented a single likelihood ratio p-value for the categorical variables (wealth index, received care). (Line 258).

Reviewer #2: 

Comment:

The presence of crepitation indicating alveolar involvement is part of the clinical diagnosis of pneumonia. It is possible that its absence in the control group may have occurred due to differential diagnoses of other respiratory diseases. I suggest removing the presence of crepitation as a risk factor.

Response: Thank you so much for your advice. We have re-analyzed the data by removing the crepitation variable as a risk factor from the multivariable model that you have suggested (Line 258).

Comment:

Regarding size, it can be as the authors did, although in our opinion it would be more appropriate to respect the case-control (1:1 ratio) study design and the cases and controls had the same number of participants

Response: Thank you so much for your suggestion. We did the secondary data analysis with all the available samples in the study. The study sample size was determined based on the original study objective. Children presenting to study clinics were assessed and those diagnosed as severe pneumonia, N=1963, and pneumonia, N=904 were included in this analysis. We kept these total estimated sample sizes and analyzed the data accordingly. Therefore, we have deleted the sample size section from the manuscript as it appeared to be unnecessary.

---

## [Decision Letter · Decision Letter 1]

14 Feb 2022

PONE-D-21-06226R1Factors associated with community acquired severe pneumonia among under-five children in Dhaka, Bangladesh: A case control analysisPLOS ONE

Dear Dr. Faruque,

Thank you for submitting your manuscript to PLOS ONE. After careful consideration, we feel that it has merit but does not fully meet PLOS ONE’s publication criteria as it currently stands. Therefore, we invite you to submit a revised version of the manuscript that addresses the points raised during the review process.

Please address the remaining comments from reviewer 1 regarding operational definitions. Please also remove reference to crepitation in the conclusion of the abstract. I think there does still need to be a section on sampling in your methods where you describe how the final sample was obtained and clarifying that you included all eligible children. Please submit your revised manuscript by Mar 31 2022 11:59PM. If you will need more time than this to complete your revisions, please reply to this message or contact the journal office at plosone@plos.org. Please include the following items when submitting your revised manuscript:A rebuttal letter that responds to each point raised by the academic editor and reviewer(s). You should upload this letter as a separate file labeled 'Response to Reviewers'.A marked-up copy of your manuscript that highlights changes made to the original version. You should upload this as a separate file labeled 'Revised Manuscript with Track Changes'.An unmarked version of your revised paper without tracked changes. You should upload this as a separate file labeled 'Manuscript'.If applicable, we recommend that you deposit your laboratory protocols in protocols.io to enhance the reproducibility of your results. Protocols.io assigns your protocol its own identifier (DOI) so that it can be cited independently in the future. For instructions see: https://journals.plos.org/plosone/s/submission-guidelines#loc-laboratory-protocols. Additionally, PLOS ONE offers an option for publishing peer-reviewed Lab Protocol articles, which describe protocols hosted on protocols.io. Read more information on sharing protocols at https://plos.org/protocols?utm_medium=editorial-email&utm_source=authorletters&utm_campaign=protocols.

We look forward to receiving your revised manuscript.

Kind regards,

Tanya Doherty, PhD

Academic Editor

PLOS ONE

Journal Requirements:

Reviewers' comments:

Reviewer's Responses to Questions

**Comments to the Author**

1. If the authors have adequately addressed your comments raised in a previous round of review and you feel that this manuscript is now acceptable for publication, you may indicate that here to bypass the “Comments to the Author” section, enter your conflict of interest statement in the “Confidential to Editor” section, and submit your "Accept" recommendation.

Reviewer #1: All comments have been addressed

2. Is the manuscript technically sound, and do the data support the conclusions?

Reviewer #1: (No Response)

3. Has the statistical analysis been performed appropriately and rigorously? 

Reviewer #1: (No Response)

4. Have the authors made all data underlying the findings in their manuscript fully available?

Reviewer #1: (No Response)

5. Is the manuscript presented in an intelligible fashion and written in standard English?

Reviewer #1: (No Response)

6. Review Comments to the Author

Reviewer #1: Please edit 'operational definitions' paragraph in lines 118 to 124 as follows for clarity: "Pneumonia was defined as a history of cough or difficult breathing and lower chest wall in-drawing or age-specific fast breathing (≥50 and ≥40 breaths/minute for 2–11 month-olds and 12–59 month-olds respectively) without any danger signs [13,14]. Severe pneumonia was defined as pneumonia with at least one of the following danger signs: central cyanosis or hypoxemia (oxygen saturation < 90% measured by pulse oximeter), severe respiratory distress (e.g. grunting, very severe chest in-drawing), inability to breastfeed or drink, lethargy or unconscious, and convulsion."

7. PLOS authors have the option to publish the peer review history of their article (what does this mean?). If published, this will include your full peer review and any attached files.

Reviewer #1: No

---

## [Author Response · Author response to Decision Letter 1]

1 Mar 2022

Comments from the Academic Editor:

Comments:

Please address the remaining comments from reviewer 1 regarding operational definitions.

Please also remove reference to crepitation in the conclusion of the abstract.

I think there does still need to be a section on sampling in your methods where you describe how the final sample was obtained and clarifying that you included all eligible children.

Response: Thank you so much for your kind review and valuable suggestions. As suggested, we have edited the ‘operational definitions’ as per reviewer 1’s suggestion in lines 126-133. 

We have removed ‘crepitation’ in the conclusion of the abstract (line 50).

We appreciate your suggestion on adding a section on sampling in method section. We have added and now it reads, “Sample size:The secondary data analysis was done with the available samples in the study. Children presenting to study clinics were assessed and those diagnosed as severe pneumonia, N=1963, and pneumonia, N=904 were included in this analysis” in lines 116-119.

Reviewer #1: 

Comment:

Reviewer #1: Please edit 'operational definitions' paragraph in lines 118 to 124 as follows for clarity: "Pneumonia was defined as a history of cough or difficult breathing and lower chest wall in-drawing or age-specific fast breathing (≥50 and ≥40 breaths/minute for 2–11 month-olds and 12–59 month-olds respectively) without any danger signs [13,14]. Severe pneumonia was defined as pneumonia with at least one of the following danger signs: central cyanosis or hypoxemia (oxygen saturation < 90% measured by pulse oximeter), severe respiratory distress (e.g. grunting, very severe chest in-drawing), inability to breastfeed or drink, lethargy or unconscious, or convulsion." 

Response: We appreciate and accept your suggestion. We have edited the ‘operational definitions’ paragraph in lines 126 to 133 as per your suggestion for clarity.

---

## [Editor Report · Decision Letter 2]

10 Mar 2022

Factors associated with community acquired severe pneumonia among under-five children in Dhaka, Bangladesh: A case control analysis

PONE-D-21-06226R2

Dear Dr. Faruque,

We’re pleased to inform you that your manuscript has been judged scientifically suitable for publication and will be formally accepted for publication once it meets all outstanding technical requirements.

Kind regards,

Tanya Doherty, PhD

Academic Editor

PLOS ONE
---

## [Editor Report · Acceptance letter]

14 Mar 2022

PONE-D-21-06226R2 

Factors associated with community acquired severe pneumonia among under five children in Dhaka, Bangladesh: A case control analysis 

Dear Dr. Faruque:

I'm pleased to inform you that your manuscript has been deemed suitable for publication in PLOS ONE. Congratulations! Your manuscript is now with our production department. 

Kind regards, 

on behalf of

Professor Tanya Doherty 

Academic Editor

PLOS ONE